# HYBRID INTERNAL MODEL: LEARNING AGILE LEGGED LOCOMOTION WITH SIMULATED ROBOT RESPONSE

**Junfeng Long**[1][*] **Zirui Wang**[1,2][*]**, Quanyi Li**[1]**, Jiawei Gao**[1,3]**, Liu Cao**[1,3]**, Jiangmiao Pang**[1][✉]

[1]OpenRobotLab, Shanghai AI Laboratory, [2]Zhejiang University, [3]Tsinghua University

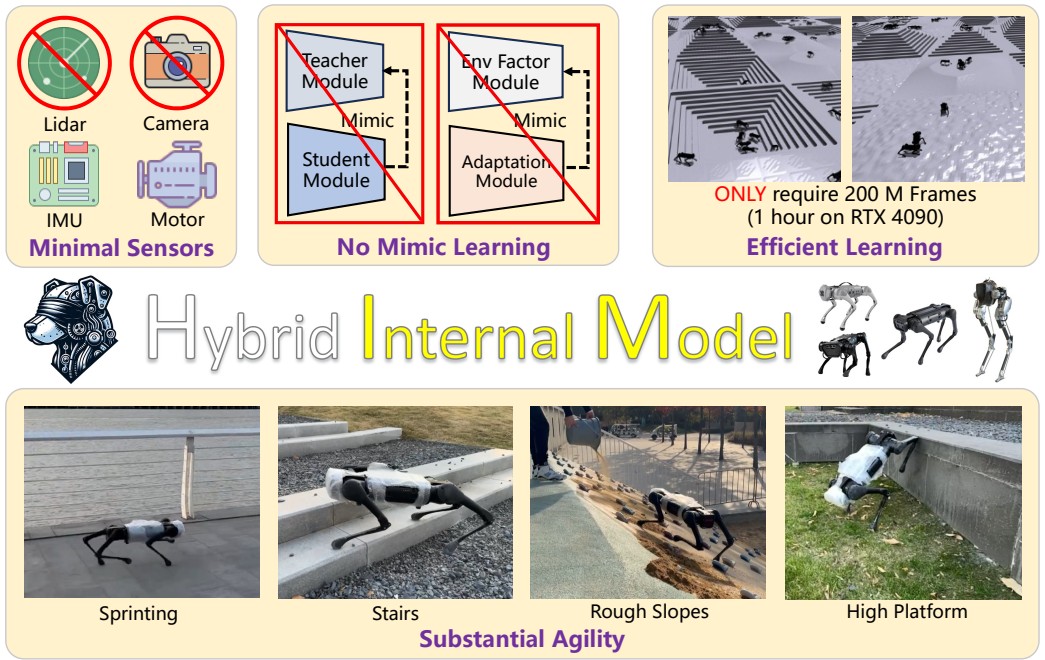

Figure 1: Our locomotion policy can drive robots to walk across any terrain under any disturbances. Key insight lies in alternatively estimating environmental dynamics with the response of the robot.

## ABSTRACT

Robust locomotion control depends on accurate state estimations. However, the sensors of most legged robots can only provide partial and noisy observations, making the estimation particularly challenging, especially for external states like terrain frictions and elevation maps. Inspired by the classical Internal Model Control principle, we consider these external states as disturbances and introduce Hybrid Internal Model (HIM) to estimate them according to the response of the robot. The response, which we refer to as the hybrid internal embedding, contains the robot's explicit velocity and implicit stability representation, corresponding to two primary goals for locomotion tasks: explicitly tracking velocity and implicitly maintaining stability. We use contrastive learning to optimize the embedding to be close to the robot's successor state, in which the response is naturally embedded. HIM has several appealing benefits: It only needs the robot's proprioceptions, i.e., those from joint encoders and IMU as observations. It innovatively maintains consistent observations between simulation reference and reality that avoids information loss in mimicking learning. It exploits batch-level information that is more robust to noises and keeps better sample efficiency. It only requires 1 hour of training on an RTX 4090 to enable a quadruped robot to traverse any terrain under any disturbances. A wealth of real-world experiments demonstrates its agility, even in high-difficulty tasks and cases never occurred during the training process, revealing remarkable open-world generalizability.

---
[*]Equal Contributions. ✉ Corresponding Author. Project page at this URL.

# 1 INTRODUCTION

In recent years, learning-based locomotion control methods for quadruped robots have achieved remarkable results (Hwangbo et al., 2019; Peng et al., 2020; Yang et al., 2020; Xie et al., 2021; Imai et al., 2022; Yu et al., 2021; Yang et al., 2022; Margolis et al., 2022; Rudin et al., 2022; Agrawal et al., 2022; Yang et al., 2023b; Margolis & Agrawal, 2023; Yang et al., 2023b), demonstrating agility beyond traditional control methods in the wild (Lee et al., 2020; Miki et al., 2022b) or on low-cost hardware platforms (Kumar et al., 2021; Wu et al., 2022). Powered by numerous and diverse simulated data, these algorithms allow the quadruped robots to work in real environments directly, and perform not only simple movements but complex skills such as manipulation (Fu et al., 2022; Ji et al., 2023; Cheng et al., 2023) and parkour (Yang et al., 2023a; Zhuang et al., 2023; Caluwaerts et al., 2023).

To achieve substantial agility, it is necessary to provide the quadruped robots with accurate and complete external states (Hartley et al., 2018). However, most onboard sensors provide partial and noisy observations about proprioception and surroundings, impeding the development of robust robot systems. As a remedy, a two-stage training strategy is usually adopted by recent learning-based methods. Firstly, an oracle policy is trained with access to external states such as terrain frictions and elevation maps in the simulator where domain randomization is applied to enhance generalizability across various terrain conditions. Subsequently, the knowledge in this policy is transferred to another policy through mimicking where external states are disabled (Miki et al., 2022a) or inferred from other information (Kumar et al., 2021). Although methods derived from this strategy yield promising results, their performance is constrained in two ways: (1) Despite the benefits of domain randomization for simulation-to-real (sim2real) transfer, the introduced noises restrict further optimization of the model, particularly when they directly learn to regress absolute environmental parameters. (2) This two-phase training paradigm complicates the training process and inevitably induces information loss in the mimicking learning phase, no matter whether it is achieved by imitation learning (Miki et al., 2022a) or adaptation methods (Kumar et al., 2021). Moreover, certain solutions require additional exteroceptive sensors, such as cameras (Agarwal et al., 2023) or lidar Rudin et al. (2022), whose configurations vary across different robot platforms. Therefore, it is challenging to train and deploy a policy on universal platforms.

In this work, we propose a method where the policy network does not require access to any external states during the entire training process. To be more specific, our method, termed Hybrid Internal Model (HIM), considers all external states such as elevation maps and ground friction as system disturbances. Inspired by the classical Internal Model Control (IMC) (Rivera et al., 1986) that simulates system response to estimate disturbances, we attempt to estimate these system disturbances according to the simulated response of the robot. Characterized by the velocity and stability, the response is termed as hybrid internal embedding, which is extracted from a sequence of historical observations. Considering the response is naturally embedded within the successor state of the robot, the hybrid internal embedding is optimized through contrastive learning, which pulls close the embedding and the future state. We use Proximal Policy Optimization (PPO) to train the policy network and feed it with both partial observations and the hybrid internal embedding. The embedding is optimized through Hybrid Internal Optimization (HIO) in each PPO iteration, allowing the policy to implicitly infer and distinguish the disturbance from the external environment. As a result, the policy can be deployed in diverse real-world settings. Also, contrastive learning makes the policy more robust to noises, and the batch-level information further improves sample efficiency. On the other hand, our method is lightweight and universal on all legged-robot platforms, as long as they provide basic proprioceptive information from IMU and joint sensors.

We train our method in Isaac Gym (Makoviychuk et al., 2021) and deploy it on Unitree Aliengo, A1, and Go1 robots. We evaluate and ablate its performance in both simulation and real-world regimes with carefully designed benchmarks and metrics. Experiments show that HIM can use minimal sensors, *i.e.*, joint encoders and IMU, to drive a robot to traverse across any terrain under any disturbances. The policy converges with only 200 million samples and only costs 1 hour on an RTX 4090 GPU for deployment. We also observe that our method can perform very well with high-difficulty tasks such as long-range stairs and the cases never occurred in the training process such as compositional terrains, and deformable slopes, revealing an excellent generalizability in the open world. We hope the simple and efficient method can bring new insights to the community.

Table 1: Comparisons between our method and previous methods. Teacher-Student refers to (Miki et al., 2022a), MONO means (Agarwal et al., 2023), AMP means Adversarial Motion Priors (Wu et al., 2022; Escontrela et al., 2022) and RMA means Rapid Motor Adaptation (Kumar et al., 2021). Current Env. Parameters indicate the elevation map, friction, restitution, *etc.* in current frame.

| Methods | Representations | External Sensors | Mimic Targets | Reference Dataset | # Samples (millions) |
|---|---|---|---|---|---|
| Teacher-Student | Current Env. Parameters | LiDAR | Behavior | - | Unknown |
| MONO | Current Env. Parameters | Depth Camera | Behavior | - | Unknown |
| AMP | Current Env. Parameters | - | Behavior | ✓ | 600 |
| RMA | Current Env. Parameters | - | Latent | - | 1,280 |
| **Ours** | **Simulated Robot Response** | - | - | - | **200** |

## 2 RELATED WORK

Historically, algorithms for legged robots have been largely rooted in traditional control-based methods, largely due to their inherent stability and robustness. Most of these works use model-based control to define controllers (Bosworth et al., 2015; Xiang et al., 2010; Yin et al., 2007; Sreenath et al., 2011; Hutter et al., 2016; Bledt et al., 2018). However, it is difficult for these controllers to adapt to situations with widely varying physical properties, such as ice, rough terrains, or deformable materials. At the same time, with the assistance of various physical simulators (Todorov et al., 2012; Panerati et al., 2021), especially the Isaac Gym (Makoviychuk et al., 2021), which enables massively parallel simulation, Deep Reinforcement Learning (DRL) for legged locomotion showcased its promising potential. This has motivated the use of learned controllers trained with RL that can adapt to changes in dynamics (Tan et al., 2018; Lee et al., 2020; Kumar et al., 2021; Fu et al., 2021; Rudin et al., 2022) in simulators, which can be deployed on the real-world robots.

While the sim2real framework exhibits attractive properties, there are gaps between simulation and real robots in two aspects. Firstly, most of the real robots cannot access external states such as elevation maps, contact forces, *etc.*. Secondly, the real-world observations are always noisy. To solve this problem, previous methods used mimic learning to compensate for the absence of environmental information. The mimic learning methods can be categorized into two main frameworks: adaptation and teacher-student. The methods using adaptation framework include Kumar et al. (2021), Agrawal et al. (2022), Margolis & Agrawal (2023) and the methods using teacher-student framework include Lee et al. (2020), Chen et al. (2019), Margolis et al. (2022), Wu et al. (2022). While these methods go some way toward solving the sim-to-real problem, there are still huge performance reductions between the reference policy and the deployable policy.

Meanwhile, there is also a different group of methods that incorporate dynamics learning to improve legged locomotion. DayDreamer (Wu et al., 2023) followed the Model-based Reinforcement learning (MBRL) framework, which uses a learned "world model" to synthesize infinite interactions; DreamWaQ (Nahrendra et al., 2023) uses a leaned representations via VAE (**?**) to boost the performance of legged locomotion. What these methods have in common is the use of a regression objective to learn a model or representation, which requires the neural network to fit the targets perfectly. However, due to the uncertainty brought by domain randomization, the network is willing to fit the random noise, eventually resulting in representation collapse.

While our method can fall into this group, a significant difference from the works above is that our method neither follows the computationally expensive MBRL framework nor uses regression objectives, which can lead to representation collapse as auxiliary losses. In contrast, we follow the IMC principle and use prototypical contrastive learning (Caron et al., 2020; Yarats et al., 2021; Deng et al., 2022) as the auxiliary loss, which only requires the encoder to distinguish how the current situation differs from other situations based on historical observations. This makes fuller use of the samples, and the sample-driven regularization leads to better robustness (LeCun, 2022).

## 3 METHODOLOGY

Our Hybrid Internal Model (HIM) can efficiently train a policy that not only promotes sim2real compatibility but also significantly enhances robotic agility. It is inspired by the classical Internal Model Control principles and implemented within a learning-based framework. Next, we first present an overview and delve into details afterward.

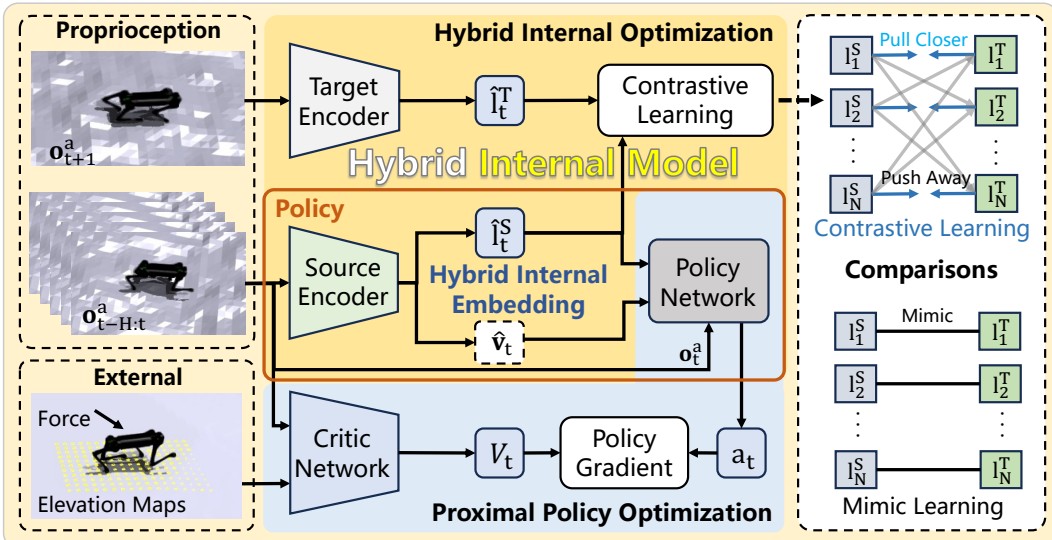

Figure 2: Overview of our framework. The policy network receives partial observations and the hybrid internal embedding, which is optimized to the robot's successor state with contrastive learning. The framework is alternatively optimized with HIO and PPO.

## 3.1 FRAMEWORK OVERVIEW

To control a legged robot, the algorithm needs to determine the movements of its actuators for a desired velocity given its proprioceptive information. We follow the paradigm (Rudin et al., 2022) that models the legged locomotion task as a sequential decision problem. The entire framework is depicted in Fig. 2. The optimization process encompasses two phases: Hybrid Internal Optimization (HIO), which trains the hybrid internal model, and Proximal Policy Optimization (PPO), which optimizes the policy network. In each iteration, we first update the parameters of HIM, then freeze them and optimize the actor and critic modules.

**State Space.** The policy network takes partial observation $\mathbf{o}_t^a$ as input, which includes the desired velocity, the proprioceptive information from its joint encoder and IMU, and the last action $\mathbf{a}_{t-1}$. The desired velocity $\mathbf{c}_t = [v_x^c, v_y^c, \omega_{\text{yaw}}^c]$ indicates the linear velocity in longitudinal and lateral directions, and the angular velocity in the horizontal direction, respectively. The joint encoder provides its joint position $\theta_t$ and joint velocity $\dot{\theta}_t$. The IMU provides its base angular velocity $\omega_t$ and gravity direction in robot frame $\mathbf{g}_t$. Our value network is allowed to access privileged information at the training stage and thus can provide a more accurate estimation of state values. Its input $\mathbf{o}_t^c$ contains two extra components to $\mathbf{o}_t^a$: current external force $\mathbf{f}_t$ and surrounding ground height $\mathbf{h}_t$.

**Action Space.** The movement of each actuator is formulated as the bias between the target joint position $\theta_{\text{target}}$ and nominal joint position $\theta_0$. To reduce the instability of the network output, we add a scale $k \leq 1$ to the policy output $\mathbf{a}_t$, thus the final target positions for joints are $\theta_{\text{target}} = \theta_0 + k\mathbf{a}_t$. The dimension of the action space $\mathcal{A}$ equals the number of actuators. For example, quadrupedal robots such as Unitree A1 or ANYmal have 12 actuators, with 3 on each leg.

**Reward Functions.** We follow the reward functions from Rudin et al. (2022) and Agarwal et al. (2023) with default weights. The details are listed in Appendix A.1.

## 3.2 HYBRID INTERNAL MODEL

The classical Internal Model Control (IMC) (Rivera et al., 1986) suggests that we can perform robust control without directly modeling the disturbance. As shown in Fig 3-(a), it uses an internal model to simulate the system response and further estimate the system disturbance, increasing the closed-loop stability. The more accurate the internal model is, the more robust control it can perform.

In legged locomotion, the external environmental dynamics such as elevation maps, ground friction, and ground restitution are disturbances to the system. However, accurately estimating them is particularly challenging. Inspired by IMC, we propose using a model to estimate the robot's response as a viable alternative. As aforementioned, the robot is explicitly given a desired velocity for locomotion. Meanwhile, it is also implicitly commanded to maintain stability. To this end, we introduce a hybrid internal model that simultaneously estimates an explicit velocity and another implicit response. These estimations, along with robot observations, are collectively fed into the policy network, forming a fully closed-loop control system. The framework is shown in Fig 3-(b).

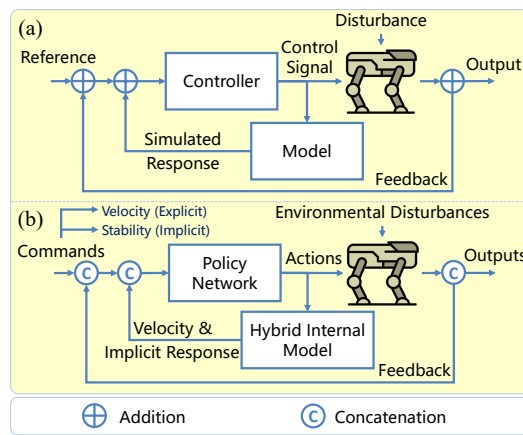

Figure 3: (a) IMC and (b) our implementations.

In practice, HIM uses sequential history observations $\mathbf{o}_{t-H:t}^a$ to extract the hybrid internal embedding that consists of the robot's velocity $\hat{\mathbf{v}}_t$ and the implicit response $\hat{\mathbf{l}}_t$. H is set as 5 by default. The extractor is a 3-layer Multi-Layer Perceptron (MLP) with hidden dimensions of 512, 256, and 128, respectively. The policy network takes the embedding and $\mathbf{o}_t^a$ as inputs and outputs actions.

### 3.3 HYBRID INTERNAL OPTIMIZATION

The most critical thing for the internal embedding $(\hat{\mathbf{v}}_t, \hat{\mathbf{l}}_t)$ is how to optimize it to simulate the robot's response, which is naturally embedded within the robot's successor state $\mathbf{o}_{t+1}^a$. In alignment with the principles of IMC, we can directly estimate $\mathbf{o}_{t+1}^a$ given $\mathbf{o}_{t-H:t}^a$. However, this is challenging due to the high dimension of robot states and the disparity between simulation and reality. Alternatively, given that we train the framework in a simulation environment, we can directly learn to regress the explicit velocity $\hat{\mathbf{v}}_t$ using its ground truth. For the implicit response $\hat{\mathbf{l}}_t$, we propose modeling it into a latent space $\mathbf{Z} \subset \mathbf{R}^{16}$ and optimizing it to be close to the successor state with contrastive learning.

In practice, in each iteration, we collect trajectories in each environment as a batch. If a pair of $\mathbf{o}_{t+1}^a$ and $\mathbf{o}_{t-H:t}^a$ belong to the same trajectory, they are positive pairs. Otherwise, they are negative pairs. The pairs are optimized by swapping assignments tasks similar to SwAV (Caron et al., 2020). Given a sequence of proprioception observations sampled from the rollout trajectories, we can derive the subsequent observation $\mathbf{o}_{t+1}^a$ from a transition as a target vector and view the concatenated historical observations $\mathbf{o}_{t-H:t}^a$ as a source vector. We ensure that the augmentation is consistent across time steps. Source and target vectors are fed to the source encoder and target encoder respectively to obtain the latent feature $\mathbf{l}_t^{\text{source}}$ and $\mathbf{l}_t^{\text{target}}$ which are mapped onto a unit sphere in high-dimensional space with a $\mathcal{L}_2$-normalization. To predict the cluster assignment probability $\mathbf{p}_t^{\text{source}}$ and $\mathbf{p}_t^{\text{target}}$ from $\mathbf{l}_t^{\text{source}}$ and $\mathbf{l}_t^{\text{target}}$, we first apply a $\mathcal{L}_2$-normalization on the prototype to obtain normalized matrix $\mathbf{E} = \{\bar{\mathbf{e}}_1, ..., \bar{\mathbf{e}}_K\}$, and then take a softmax over the dot products of source vectors or target vectors with all the prototypes:

$$\mathbf{p}_t^{\text{source}} = \frac{\exp(\frac{1}{\tau}\mathbf{l}_t^{\text{source}\top}\mathbf{e}_k)}{\sum_{k'}\exp(\frac{1}{\tau}\mathbf{l}_t^{\text{source}\top}\mathbf{e}_{k'})}, \quad \mathbf{p}_t^{\text{target}} = \frac{\exp(\frac{1}{\tau}\mathbf{l}_t^{\text{target}\top}\mathbf{e}_k)}{\sum_{k'}\exp(\frac{1}{\tau}\mathbf{l}_t^{\text{target}\top}\mathbf{e}_{k'})}. \quad (1)$$

Here, $\mathbf{p}_t^{\text{source}}$ and $\mathbf{p}_t^{\text{target}}$ are the predicted probability that historical observations $\mathbf{o}_{t-H:t}^a$ maps to individual cluster with index $k$, while $\tau$ is a temperature parameter.

To obtain the targets $(\mathbf{q}_1^{\text{source}}, ..., \mathbf{q}_K^{\text{source}})$ and $(\mathbf{q}_1^{\text{target}}, ..., \mathbf{q}_K^{\text{target}})$ for the aforementioned predicted probabilities, while avoiding trivial solutions, the Sinkhorn-Knopp algorithm (Cuturi, 2013) is applied to both encoders. Now that we have the cluster assignment predictions and targets, the representation learning objective is simply to maximize the prediction accuracy:

$$\mathcal{J}^{\text{SwAV}} = -\frac{1}{2H}\sum_{t=1}^{H}(\mathbf{q}_t^{\text{source}}\log\mathbf{p}_t^{\text{target}} + \mathbf{q}_t^{\text{target}}\log\mathbf{p}_t^{\text{source}}). \quad (2)$$

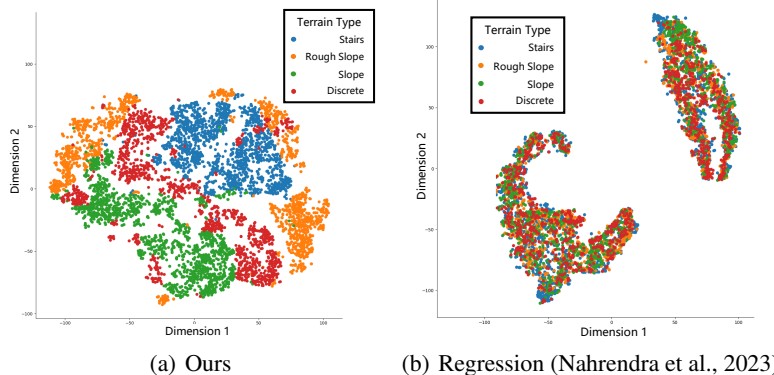

(a) Ours           (b) Regression (Nahrendra et al., 2023)

Figure 4: Latent space visualizations of (a) our hybrid internal model and (b) Nahrendra et al. (2023).

**Analysis.** Our method can maximize the similarity of latent features between historical observations and the next observations, implicitly modeling external states without requirements for regression. This also improves the performance of locomotion policy by making use of the information in batch level, *i.e.*, the different environment properties in different kinds of terrain. We conduct t-SNE Van der Maaten & Hinton (2008) on the latent outputs of our methods and regression (Nahrendra et al., 2023). The visualization shows that our hybrid internal model has a more separable encoding of environments, which means that our method carries more precious environmental information, resulting in a stronger ability to identify which kind of terrain the robot is on.

## 3.4 TRAINING DETAILS

**Simulation Setup.** We use Isaac Gym (Rudin et al., 2022) with 4096 parallel environments and a rollout length of 100 time steps. The training process needs 1000 rollouts which takes 1 hour of wall clock time on NVIDIA RTX 4090. But its performance continues improving until 2000 rollouts.

**Dynamics Randomization.** To improve the robustness of the policy and facilitate the sim-to-real process, we randomize the mass of the robot body and links, the centre of mass (CoM) of the robot, the payload applied to the body of the robot, the ground friction and restitution coefficients, the motor strength, the joint-level PD gains, the system delay, the external force, and the initial joint positions in each episode. The randomization ranges for each parameter are detailed in Appendix A.2.

**Training Curriculum.** We use a similar terrain curriculum as Rudin et al. (2022) and Wu et al. (2022). We create a height field map with 200 terrains arranged in a $20 \times 10$ grid, and each row has the same type of terrain arranged in increasing difficulty, while each grid is $10 \times 10 \, \text{m}^2$. The detailed description of our training terrain is in Appendix A.3. We put all the robots in different types of terrains with the lowest difficulty at the beginning, the terrain level will increase if the robot reaches 80% of linear tracking reward, and will decrease if they can not travel half of the terrains in one episode. We use a simple command curriculum to help the robots learn from easy to hard. On complex terrains such as stairs or discrete obstacles, we sample longitudinal and lateral linear velocity commands in $[-1.0, \, 1.0]$ m/s and horizontal angular velocity in $[-2.0, \, 2.0]$ rad/s. On slopes and rough slopes, we increase the range of longitudinal linear velocity to $[-3.0, \, 3.0]$ m/s and the range of horizontal angular velocity to $[-3.0, \, 3.0]$ rad/s. The commands are all sampled uniformly in the corresponding range and independently for each robot, we sample the commands every 25 time steps.

## 4 EXPERIMENTS

In this section, we evaluate the performance of our method in both simulation and real-world regimes and conduct ablation studies. The demo videos of our method can be found at the Project Page. Additional descriptions of the experimental setup, baselines, and hyper-parameters can be found in the Appendix.

Table 2: Main results with average performance ↑ in the real-world benchmarks over 20 trials, where ± captures a 95% confidence interval. The top results are highlighted.

| Benchmarks | Environments | Metrics | **Ours** | RMA | MoB | Built-in MPC |
|---|---|---|---|---|---|---|
| Stairs | Short-range (A1) | Success rate (%) | 100 | 60 | 0 | 0 |
| | Long-range (Aliengo) | Number of stairs | 176.5±7.81 | 75.35±19.98 | 0.0±0.0 | 0.0±0.0 |
| Unseen Terrains | Compositional Terrain (Aliengo) | Success rate (%) | 85 | 45 | 0 | 0 |
| | Deformable Slope (A1) | Success rate (%) | 55 | 10 | 0 | 0 |
| Anti-disturbance | Dragging Obstacle (A1) | Maximum weight (Kg) | 10 | 10 | 7 | 3 |
| | Vertical Hit (A1) | Maximum weight (Kg) | 8 | 7.5 | 7 | 5 |
| | Payload (A1) | Maximum weight (Kg) | 8 | 7 | 4 | 7 |
| | Missing steps (Aliengo) | Success rate (%) | 100 | 0 | 0 | 0 |

## 4.1 EVALUATION SETUPS

**Compared Methods.** We compare our method with the following methods:
• Baseline: Train without HIM but keep the source encoder and optimize it with PPO.
• Ours w/o velocity input: Set the velocity inputs to the policy network as 0.
• Ours w/o velocity loss: Direct remove the velocity estimation.
• Ours w/o internal latent input: Set the internal latent inputs to the policy network as 0.
• Ours w/o internal latent loss: Direct remove the internal latent estimation.
• Regression: Optimize the hybrid internal model with regression methods.
• Oracle: Train the policy with a history of full observations.
• Rapid Motor Adaptation (Kumar et al., 2021).
• Multiplicity of Behavior (Walk These Ways) (Margolis & Agrawal, 2023).
• Onboard MPC controller, which is only compared in real-world experiments.

**Setups in the Real World.** We deploy our policy on Unitree A1, Go1, and Aliengo. The policy runs at 50 Hz, with a PD controller running at 500 Hz to track the target. The parameters are $k_p = 30.0$, $k_d = 0.75$ for A1 and Go1, $k_p = 40.0$, $k_d = 2.0$ for Aliengo. We compare methods with robust tests in the real world, with 3 benchmarks and 8 tasks. The details can be found in Appendix B.1.

**Setups in Simulations.** In simulations, we compare methods according to the tracking errors. The tracking errors for linear velocity and angular velocity are quantified using the norms $\|v_{x,y} - v_{x,y}^{\text{target}}\|_2$ and $\|\omega_{\text{yaw}} - \omega_{\text{yaw}}^{\text{target}}\|_2$, respectively. We test the tracking performance under various terrain conditions and motion ranges. A detailed description can be found in Appendix B.2.

## 4.2 MAIN RESULTS

**Real-world Results** In our real-world benchmarks, we compare our method with RMA (Kumar et al., 2021), MoB (Margolis & Agrawal, 2023), and Built-in MPC. The results in Table 2 show our method outperforms the other methods on all tasks and maintains more natural gaits. It proves that our method can well inherit the policy in simulation and can be deployed to the real world easily. We also observe that our method can perform very well with high-difficulty tasks such as long-range stairs and the cases never occurred in the training process such as compositional terrains, and deformable slopes, revealing an excellent generalizability in the open world.

**Simulation Results** In our simulation benchmarks, we compare our method with Baseline, Regression, RMA (Kumar et al., 2021) and MoB (Margolis & Agrawal, 2023) with Unitree Aliengo. Results in Table 3 show that our method outperforms the other methods on almost all tasks. Our method holds more performance improvements than the other methods and significantly outper-

Table 3: Average tracking error ↓ in simulation over $4096 \times 5$ trials. Top results are highlighted.

| Terrain Types | Velocity Types | Ranges | **Ours** | Regression | RMA | Baseline | MoB (Trot) |
|---|---|---|---|---|---|---|---|
| Slopes | Linear | [-1, 1] m/s | 0.073 | 0.071 | 0.158 | 0.271 | 0.103 |
| | | [-2, 2] m/s | 0.075 | 0.077 | 0.270 | 0.305 | 0.126 |
| | Angular | [-1, 1] rad/s | 0.084 | 0.086 | 0.091 | 0.164 | 0.081 |
| | | [-2, 2] rad/s | 0.124 | 0.130 | 0.134 | 0.202 | 0.114 |
| | Combined | [-1, 1] m/s [-1, 1] rad/s | 0.078 0.047 | 0.075 0.050 | 0.155 0.074 | 0.207 0.143 | 0.069 0.052 |
| | | [-2, 2] m/s [-2, 2] rad/s | 0.094 0.067 | 0.107 0.071 | 0.203 0.102 | 0.224 0.157 | 0.103 0.069 |
| Rough Slopes | Linear | [-1, 1] m/s | 0.086 | 0.091 | 0.201 | 0.287 | 0.105 |
| | | [-2, 2] m/s | 0.085 | 0.088 | 0.294 | 0.331 | 0.124 |
| | Angular | [-1, 1] rad/s | 0.097 | 0.108 | 0.142 | 0.160 | 0.119 |
| | | [-2, 2] rad/s | 0.143 | 0.150 | 0.147 | 0.187 | 0.135 |
| | Combined | [-1, 1] m/s [-1, 1] rad/s | 0.088 0.054 | 0.095 0.070 | 0.167 0.086 | 0.220 0.169 | 0.147 0.092 |
| | | [-2, 2] m/s [-2, 2] rad/s | 0.103 0.076 | 0.137 0.096 | 0.221 0.117 | 0.284 0.201 | 0.195 0.114 |
| Stairs | Linear | [-1, 1] m/s | 0.147 | 0.194 | 0.306 | 0.377 | 1.108 |
| | Angular | [-1, 1] rad/s | 0.087 | 0.101 | 0.241 | 0.370 | 0.930 |
| | Combined | [-1, 1] m/s [-1, 1] rad/s | 0.135 0.096 | 0.217 0.154 | 0.337 0.201 | 0.405 0.289 | 1.270 0.992 |
| Discretre Obstacles | Linear | [-1, 1] m/s | 0.125 | 0.182 | 0.311 | 0.279 | 0.973 |
| | Angular | [-1, 1] rad/s | 0.083 | 0.097 | 0.175 | 0.253 | 0.841 |
| | Combined | [-1, 1] m/s [-1, 1] rad/s | 0.125 0.075 | 0.206 0.133 | 0.305 0.177 | 0.374 0.246 | 1.254 0.976 |

forms the Regression method. Besides, our method also performs better in 66.67% of the velocity range on easy terrain.

## 4.3 ABLATION STUDIES

**Ablation Studies in Simulations.** We mainly conduct ablation studies in simulation with Aliengo. The metrics include the normalized linear velocity tracking score, the normalized angular velocity tracking score, and the maximum reachable terrain level. The task and command sampling are described in Section 3.4. The definition of terrain levels can be found in Appendix A.3. Terrains include rough flats, pyramid slopes, rough pyramid slopes, wave terrain, stairs, and flats with discrete obstacles, and the corresponding proportion is [0.1, 0.2, 0.6, 0.1]. The normalized linear velocity tracking score (NLTS) and the normalized angular velocity tracking score (NATS) are:

$$\text{NLTS} = \exp(-\frac{\|v_{x,y} - v_{x,y}^{\text{target}}\|_2^2}{0.25}), \ \text{NATS} = \exp(-\frac{\|\omega_{\text{yaw}} - \omega_{\text{yaw}}^{\text{target}}\|_2^2}{0.25}).$$

The results are shown in Fig. 4.3, in which the curves are averaged over 10 seeds. The shaded area represents the standard deviation across seeds. Despite our method does not need terrain information, it exhibits the most similar results to the Oracle policy. We also conclude that:

1. In our hybrid internal model, both internal embedding and velocity estimation improve the policy. The absence of any of them can lead to performance degradation.
2. Our hybrid internal optimization outperforms the Regression method. It proves that our method can utilize batch-level information to better understand the properties of different terrains.
3. RMA can not imitate all the great performances of Oracle. Our method maintains consistent observations between simulation and reality and results in better performance.

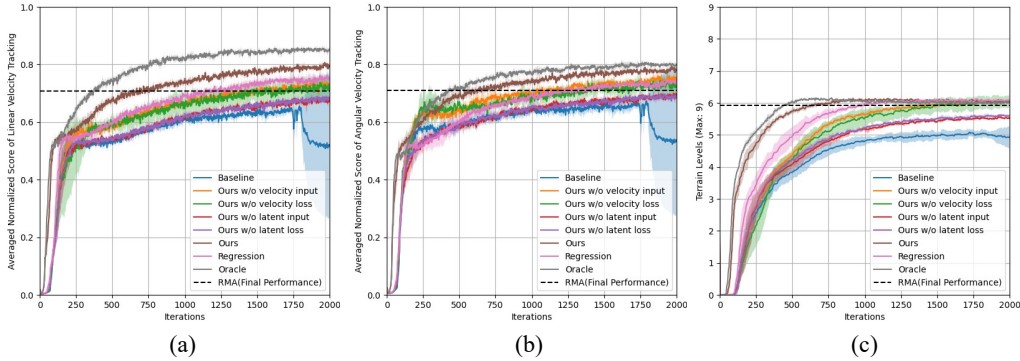

Figure 5: Ablation studies with learning curves of (a) normalized linear velocity tracking score, (b) normalized angular velocity tracking score, and (c) maximum reachable terrain level in Isaac Gym.

Table 4: Ablation studies with average performance ↑ in the real-world benchmarks over 20 trials, where ± captures a 95% confidence interval. Top performances are highlighted.

| Benchmarks | Environments | Metrics | **Ours** | Ours w/o vel. inp. | Ours w/o vel. loss | Ours w/o lat. inp. | Ours w/o lat. loss | Regression | Baseline |
|---|---|---|---|---|---|---|---|---|---|
| Stairs | Short-range (A1) | Success rate (%) | 100 | 85 | 80 | 50 | 55 | 85 | 10 |
| | Long-range (Aliengo) | Number of stairs | 176.5±7.81 | 51.45±25.67 | 54.6±24.68 | 13.3±13.42 | 10.6±9.68 | 155.7±13.48 | 0.0±0.0 |
| Unseen Terrains | Compositional Terrain (Aliengo) | Success rate (%) | 85 | 70 | 75 | 50 | 50 | 70 | 15 |
| | Deformable Slope (A1) | Success rate (%) | 55 | 30 | 25 | 0 | 0 | 30 | 0 |
| Anti-disturbance | Dragging Obstacle (A1) | Maximum weight (Kg) | 10 | 10 | 10 | 5 | 5 | 10 | 4 |
| | Vertical Hit (A1) | Maximum weight (Kg) | 8 | 7 | 6.5 | 4 | 4 | 8 | 5.5 |
| | Payload (A1) | Maximum weight (Kg) | 8 | 7 | 7 | 3.5 | 3.5 | 7.5 | 3 |
| | Missing steps (Aliengo) | Success rate (%) | 100 | 90 | 90 | 5 | 0 | 65 | 0 |

**Ablation Studies in the Real World.** We also conduct ablation studies on our real-world benchmarks. Results in Table 4 show that our hybrid internal model contributes to the performance, with the internal model playing a critical role.

## 5 CONCLUSION

Our work presents a promising framework and reveals the connection between learning-based locomotion control and classical Internal Model Control principles. In addition, our experimental results show that joint encoders and an IMU provide sufficient information. We also demonstrate that compared to regression, contrastive learning provides a better representation in terms of robustness to noise and high sample efficiency. To the best of our knowledge, our method is the first system to fully utilize batch-level information in locomotion tasks, leveraging massively parallel simulations of Isaac Gym. However, the lack of external perceptions, e.g., visual information, leaves legged robots with only a primitive skill set. We will incorporate our method with external perception in the future to overcome more challenging environments and higher-level tasks.

**Acknowledgement.** This work is supported by Shanghai Artificial Intelligence Laboratory.

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

# A HYPER-PARAMETERS

## A.1 REWARD FUNCTIONS

The reward functions we used during the training are shown in Table 5, which come from Kumar et al. (2021); Agarwal et al. (2023); Margolis & Agrawal (2023); Fu et al. (2021).

Table 5: Rewards

| Reward | Equation $(r_i)$ | Weight $(w_i)$ |
|---|---|---|
| Lin. velocity tracking | $\exp\left\{-\frac{\|\mathbf{v}_{xy}^{\mathrm{cmd}}-\mathbf{v}_{xy}\|_2^2}{\sigma}\right\}$ | 1.0 |
| Ang. velocity tracking | $\exp\left\{-\frac{\left(\omega_{\mathrm{yaw}}^{\mathrm{cmd}}-\omega_{\mathrm{yaw}}\right)^2}{\sigma}\right\}$ | 0.5 |
| Linear velocity $(z)$ | $v_z^2$ | -2.0 |
| Angular velocity $(xy)$ | $\|\boldsymbol{\omega}_{xy}\|_2^2$ | -0.05 |
| Orientation | $\|\mathbf{g}\|_2^2$ | -0.2 |
| Joint accelerations | $\|\ddot{\boldsymbol{\theta}}\|^2$ | $-2.5 \times 10^{-7}$ |
| Joint power | $\|\boldsymbol{\tau}\|\|\dot{\boldsymbol{\theta}}\|^T$ | $-2 \times 10^{-5}$ |
| Body height | $(h^{\mathrm{target}} - h)^2$ | -1.0 |
| Foot clearance | $\sum_{i=0}^{3}\left(p_z^{\mathrm{target}} - p_z^i\right)^2 \cdot v_{xy}^i$ | -0.01 |
| Action rate | $\|\mathbf{a}_t - \mathbf{a}_{t-1}\|_2^2$ | -0.01 |
| Smoothness | $\|\mathbf{a}_t - 2\mathbf{a}_{t-1} + \mathbf{a}_{t-2}\|_2^2$ | -0.01 |

In Table 5, $\sigma$ is the tracking shaping scale and we use $\sigma = 0.25$, $h^{\mathrm{target}}$ is the desired base height corresponding to ground, $p_z^{\mathrm{target}}$ and $p_z^i$ are the desired feet position and real feet position in z-axis of robot's frame and $v_{xy}^i$ is the feet velocity in xy-plane of robot's frame.

## A.2 DOMAIN RANDOMIZATIONS

For the training of our method and other methods, domain randomizations we use are shown in Table 6, which come from Wu et al. (2022); Nahrendra et al. (2023).

Table 6: Domain Randomizations and their Respective Range

| Parameters | Range[Min, Max] | Unit |
|---|---|---|
| Body Mass | $[0.8, 1.2] \times$ nominal value | Kg |
| Link Mass | $[0.8, 1.2] \times$ nominal value | Kg |
| CoM | $[-0.1, 0.1] \times [-0.1, 0.1] \times [-0.1, 0.1]$ | m |
| Payload Mass | $[-1, 3]$ | Kg |
| Ground Friction | $[0.2, 2.75]$ | - |
| Ground Restitution | $[0.0, 1.0]$ | - |
| Motor Strength | $[0.8, 1.2] \times$ motor torque | Nm |
| Joint $K_p$ | $[0.8, 1.2] \times 20$ | - |
| Joint $K_d$ | $[0.8, 1.2] \times 0.5$ | - |
| Initial Joint Positions | $[0.5, 1.5] \times$ nominal value | rad |
| System Delay | $[0, 3\Delta_t]$ | s |
| External Force | $[-30, 30] \times [-30, 30] \times [-30, 30]$ | N |

## A.3 TRAINING TERRAINS

Our terrain comprises slopes, rough slopes, stairs, and discrete obstacles.

The slopes and rough slopes vary in inclination from $0°$ to $40°$ with the addition of uniform noise ranging from $\pm 1$ cm to $\pm 8$ cm on the rough slopes. The stairs have random widths ranging from 20 cm to 40 cm and heights varying from 5 cm to 23 cm. The discrete obstacles have heights ranging from $\pm 5$ cm to $\pm 15$ cm. We assign difficulty levels to the terrains using integers from 0 to 9. More specifically, for a terrain level indicated by $\text{level}^{\text{terrain}}$, the corresponding terrains are as follows:

- Slopes: with inclination of $40 \times \frac{\text{level}^{\text{terrain}}}{9}$ $°$;
- Rough slopes: with amplitude of $1 + 7 \times \frac{\text{level}^{\text{terrain}}}{9}$ cm and inclination of $40 \times \frac{\text{level}^{\text{terrain}}}{9}$ $°$;
- Stairs: with step height of $5 + 18 \times \frac{\text{level}^{\text{terrain}}}{9}$ cm;
- Discrete obstacles: with height of $5 + 10 \times \frac{\text{level}^{\text{terrain}}}{9}$ cm. The proportion of the above terrains is 0.1, 0.2, 0.6, and 0.1 correspondingly.

## A.4 HYPER PARAMETERS

Table 7: Hyper Parameters for Training

| General | |
| --- | --- |
| Batch Size | $4096 \times 200$ |
| Mini-batch Size | $4096 \times 50$ |
| Number of epochs | 5 |
| **PPO** | |
| Clip range | 0.2 |
| Entropy coefficient | 0.01 |
| Discount factor | 0.99 |
| GAE discount factor | 0.95 |
| Desired KL-divergence | 0.01 |
| Learning rate | $1 \times 10^{-3}$ |
| Adam epsilon | $1 \times 10^{-8}$ |
| Gradient clipping | 10 |
| **HIM** | |
| Number of prototypes | 16 |
| Contrastive loss scale | 1.0 |
| Velocity estimation loss scale | 1.0 |
| Learning rate | $1 \times 10^{-3}$ |
| Adam epsilon | $1 \times 10^{-8}$ |
| Gradient clipping | 10 |

# B  EXPERIMENTAL DETAILS

## B.1  REAL-WORLD BENCHMARK SETUP

Our benchmark consists of three parts: **stairs**, **unseen terrains** and **anti-disturbance**:

**Stairs:**
• Short-range stairs: This scene includes two sets of stairs with 5 steps each. The height and width of the stairs are 16 cm and 27 cm, respectively. The Unitree A1 robot is used to test our policy in this scene, and the metric is the success rate of going through both stairs. The corresponding snapshots of this scene are displayed in the left column of Fig. 6(a).
• Long-range stairs: This scene consists of hundreds of stairs with a height of 17 cm and a width of 25 cm. The Unitree Aliengo robot is equipped with policies that need to be tested, and the metric is the number of stairs the robot can ascend without stopping. The corresponding photos of this scene are shown in the right column of Fig. 6(a).

**Unseen terrains:**
• Compositional terrain: This scene alternates between gravel and stairs, as depicted in the upper picture of Fig. 6(b). The Unitree Aliengo robot is equipped with policies that need to be tested, and the metric is the success rate of going through three compositional alternations.
• Deformable slope: This scene makes a deformable slope created by two soft beds, as shown in the lower picture of Fig. 6(b). The Unitree A1 robot is used to test our policy, and the metric is the success rate of navigating through this slope.

**Anti-disturbance:**
• Dragging obstacle: This task simulates the scenario where the robot's leg is dragged by obstacles. A wooden box is attached to the robot's back leg, preventing its movement. The metric for this task is the maximum weight the robot can handle. The Unitree A1 robot is used, and the illustration can be found in the top-left of Fig. 6(c).
• Lateral hit: This task evaluates the robot's push-recovery ability. The robot is subjected to a lateral force from a load swinging down from a height of 107 cm. The Unitree A1 robot is used in this task, and the metric is the maximum weight of the load that the robot can withstand after being hit. The illustration is shown in the top-right of Fig. 6(c).
• Payload: This task assesses the robot's carrying capacity. The Unitree A1 robot is used, and the maximum weight that the robot can carry while walking is measured. The illustration can be found in the bottom-left of Fig. 6(c).
• Missing step: This task simulates the robot's reaction when it misses steps and falls from a high platform. The platform's height, as shown in the bottom-right of Fig. 6(c), is 42 cm. The Unitree Aliengo robot is used in this task, and the metric is the success rate of maintaining balance after descending.

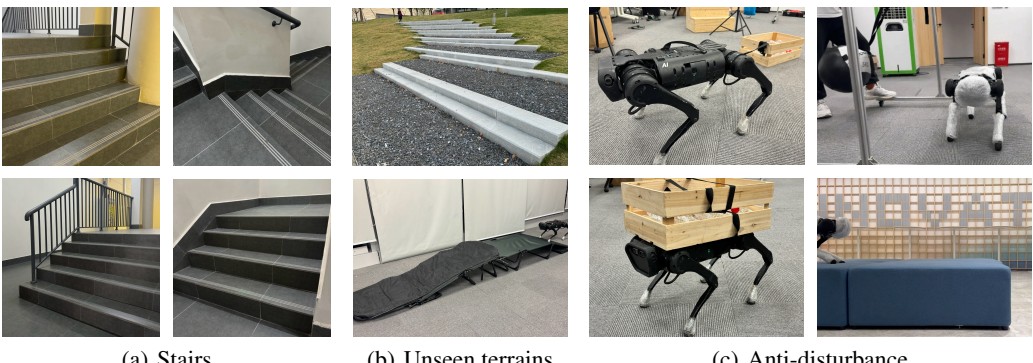

|     (a) Stairs     |     (b) Unseen terrains     |     (c) Anti-disturbance     |

Figure 6: Snapshots of our real-world scene benchmarks for quantitative analysis in Section 4.1.

## B.2 SIMULATION BENCHMARK SETUPS

We consider tracking tasks with 3 different ranges of commands for each type of terrain. The tracking error on slopes and rough slopes are tested in the range $[-1.0, 1.0]$ m/s × $[-1.0, 1.0]$ m/s × $[-1.0, 1.0]$ rad/s and $[-2.0, 2.0]$ m/s × $[-1.0, 1.0]$ m/s × $[-2.0, 2.0]$ rads/s. All other tests are conducted within the range $[-1.0, 1.0]$ m/s × $[-1.0, 1.0]$ m/s × $[-1.0, 1.0]$ rad/s. These ranges correspond to linear velocity in the forward direction, lateral directions, and angular velocity about the vertical axis, respectively.

When testing the linear velocity tracking error, the angular velocity target is set to 0. For the angular velocity tracking error tests, the linear velocity target is 0. The 'combined velocity' tests are conducted without setting either velocity component to 0. We uniformly sample test commands within these ranges. The environmental conditions are set to $\text{level}^{\text{terrain}} = 1, 2, 3, 4$, with an equal distribution of 25% for each, as explained in Appendix A.3.

## B.3 ABLATION ON NUMBER OF PROTOTYPES

We conducted ablation studies on the number of prototypes in our research. By default, we set the number of prototypes to 16, which corresponds to the square of terrain types.

The training curves for different numbers of prototypes are shown in Fig. 7. It can be observed that the performance of our method steadily improves as the number of prototypes increases until it reaches 64. After this, the performance starts to decline.

This finding suggests that the representation of environmental dynamics does not require dense representation like pure regression, but overly sparse representation is also inadequate. Additionally, a high number of prototypes accelerates training during the initial stages but compromises final performance. Conversely, a low number of prototypes may lead to slower initial training but ultimately achieves superior final performance.

Thus, determining an appropriate number of prototypes is vital for striking a balance between rapid training speed and optimal final performance. This matter warrants further investigation in future studies.

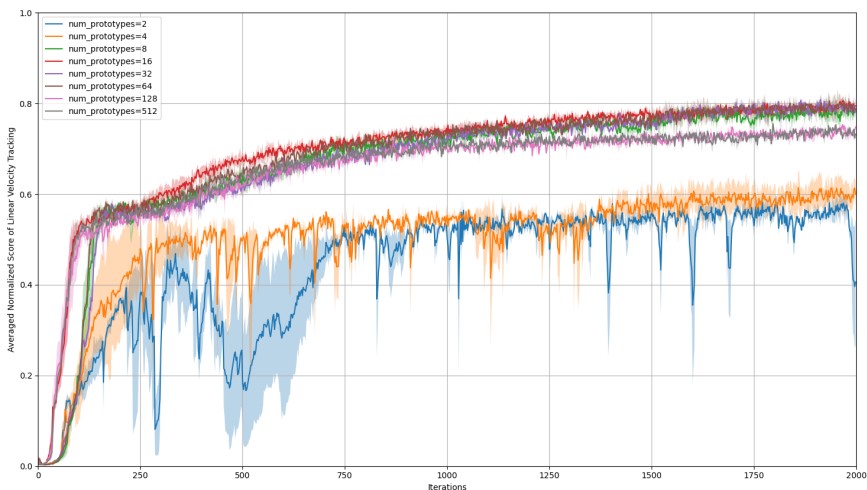

Figure 7: Normalized Linear Tracking Score over the number of Prototypes

## B.4 OUTDOOR EXPERIMENTS

We further evaluated our policy in additional real-world scenes. Videos can be found at the following URL. Illustrations of the scenes that have appeared during our outdoor experiments are shown in Fig. 8.

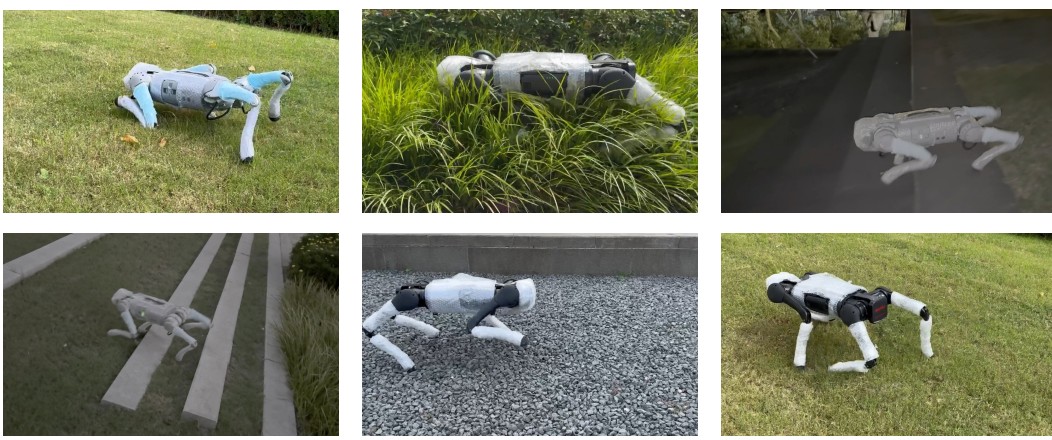

Figure 8: Snapshots of various outdoor scenes for the agility evaluation of our quadruped robot.

