# OpenReview forum: "Hybrid Internal Model: Learning Agile Legged Locomotion with Simulated Robot Response"
_ICLR.cc/2024/Conference — ICLR 2024 poster_

### Official Review · Reviewer_9L23 · 2023-10-29

**Soundness:** 1 poor
**Presentation:** 1 poor
**Contribution:** 2 fair
**Rating:** 5
**Confidence:** 5

**Summary:**

This work proposees Hybrid Internal Model which uses minimal sensor input for legged locomotion aiming to address the limitation of existing learning-based locomotion control methods. The Hybrid Internal Model learns the latent representation of velocity and implicit dynamics of the environment via contrastive learning. The method is evaluated in simulation and deployed in the real world.

**Strengths:**

1. The work provides a simple one-stage method for training blind locomotion controllers, with minimal sensory information.
2. This work provides impressive real-world performance in the supplementary video.
3. The t-SNE visualization shows that the proposed hybrid internal model learned to distinguish different terrain type in latent space.

**Weaknesses:**

1. The overall delivery of the work is not satisfactory in the following aspect:
* Many technical details are poorly described or totally missing: what’s the exact task legged robot need to solve (follow forward velocity or follow angular velocity or anything else), what command is included in the user-command, what’s the linear velocity and angular velocity range for the input.
* Some parts are quite confusing, for instance: the user command and trainable prototypes are using the same notation: c_{i}. There are many typos in the paper, for example: “police network” in section 3.

2. The evaluation of the method is poor:
* There is no quantitative comparison with any baseline in the paper (no result in simulation and no result in real world).
* The only comparison result (The training curve in Figure 6) provides limited information, since there is no specific description of the task. It’s hard to see the actual performance gap of different methods in terms of the “task target” like velocity tracking and angular.0
* There is no actual baseline compared in the experiment section, since the method use only proprioception observation of the robot, at least RMA, MoB should be compared, quantitatively, and provide corresponding training curve. Since this work claim the proposed method provide better performance and sample efficiency against two-stage methods, at least comparison is needed.
* There is no performance analysis the method or baselines across different terrains, or across different command range or different command types, or across different command target.
* This work claims contrastive learning is better than purely regression, but there is no experiment to support the claim.


Reference:
[1] Kumar, et al. RMA: Rapid Motor Adaptation for Legged Robots

**Questions:**

Though this work provides a reasonable real-world demo, this work need to be improved in multiple ways to meet the standard of ICLR.
1. Many technical details (including but not limited to the items mentioned in the weakness section) need to be cleared.
2. More baseline needs to be compared.
3. More ablation study need to be performed, for example regression or contrastive learning
4. More specific performance analysis is required: performance across terrain or different physical properties.

---

> ### Author Response · Authors · 2023-11-17
> **Response to Reviewer 9L23**
>
> # Response to Reviewer 9L23
>
> ## Comment
> Thank you for your review and comments. To respond/comment on your listed weaknesses:
>
> W1: We have fixed the typos in our paper. The task is to tracking the user command including linear velocity in x-axis, linear velocity in y-axis and yaw angular velocity, The range differs from terrain type, for example, the command of linear velocity in x-axis on slopes and rough slopes is [-3.0, 3.0] m/s and [-1.0, 1.0] m/s on stairs, the range of linear velocity in y-axis is [-1.0, 1.0] m/s on all terrains, the range of yaw angular velocity on slopes and rough slopes is [-3.0, 3.0] rad/s and [-1.0, 1.0] rad/s on stairs. We also update a more clear decription of technical details in $\color{red}{section 3}$ in current version.
>
> W2: We have done more evaluation of our method.
> * We conduct quantitative comparison with various baselines in both simulation and real world in $\color{red}{section 4.3}$ and $\color{red}{section 4.4}$ in current version.
>
> * We already clearify the task description of both simulation and real world benchmarks. In simulation benchmarks, we test the trackng error across various terrains and command ranges. In real world benchmarks, we set up three types of terrains to evaluate the performance of real world deployment, including success rate, number of stairs, maximum weights of payload, and other metrics. You can checkout the details in $\color{red}{section 4.3}$ and $\color{red}{section 4.4}$ in current version.
>
> * We compared our method with the baselines including the regression method, RMA, MoB, Built-in MPC and ours w/o auxiliary loss, the experimental results show that our method outperform other baselines in both simulation and real-world. The experimental details are in $\color{red}{section 4.3}$ and $\color{red}{section 4.4}$.
>
> * We conduct performance analysis of our method and other baselines across different terrains, command ranges and command types. For our tasks, the only target is to track the velocity commands. You can checkout the details in $\color{red}{section 4.3}$ and $\color{red}{section 4.4}$ in current version.
>
> * We conduct extensive ablation studies on  contrastive learning and regression. To be more specificly, we perform ablation study with the following methods:
>     * **Baseline**: Training the policy with no auxiliary task, but with a history encoder of the same architecture training as our method, the history encoder is trained with policy update.
>     * **Ours w/o Velocity Input**: Training the policy the same as our method, but set the position of velocity input to 0 for the policy network.
>     * **Ours w/o Velocity Loss**: Training the policy the same as our method, but without the loss of velocity estimation and set the position of velocity input to 0 for the policy network.
>     * **Ours w/o Latent Input**: Training the policy the same as our method, but set the position of latent input to 0 for the policy network.
>     * **Ours w/o Latent Loss**: Training the policy the same as our method, but without the loss of dynamics-aware estimation and set the position of latent input to 0 for the policy network.
>     * **Regression**: Training policy with velocity estimation and dynamics-aware estimation, but the auxiliary is to simply regress them in latent space.
>     * **Oracle**: Training policy with a history of full observations.
>
>     The experiments include simulation and real-world, showing that our method reaches good performance across simulation and real-world. You can see $\color{red}{section 4.5}$ for detail information.
>
> ## Questions
> Q1: We answered this question in W1. A more clear decription of technical details can be found in $\color{red}{section 3}$.
>
> Q2: We answered this question in the point 1,2,3 of W2. We compared our method with the following baselines including the regression method, RMA, MoB, Built-in MPC and ours w/o auxiliary
> task, the result shows that our method ou-perform in both simulation and real-world, the experiments are in $\color{red}{section 4.3}$ and $\color{red}{section 4.4}$.
>
> Q3:  We answered this question in the point 5 of W2. We perform ablation studies with various methods. The experiments include simulation and real-world, showing that our method reaches good performance across simulation and real-world. You can checkout $\color{red}{section 4.5}$ for detail information.
>
> Q4: We answered this question in the point 4 of W2. Our new experiments in $\color{red}{section 4.3}$ and $\color{red}{section 4.4}$ include the performance of our method. In simulator, we test the tracking error over different terrains in different command ranges; in real world we setup a benchmark including stairs, unseen terrains and anti-disturbance, our method also out-performs in almost all the task.

---

> ### Author Response · Authors · 2023-11-21
> **Extension of Experiment Part(Comparison part 1)**
>
> In order to compare our method with other blind locomotion method, we conduct a tracking error($\color{red}{the \ Euclidean \ distance \ between \ real \ velocity \ and \ target \ velocity}$) benchmark in simulation. The tracking error on slopes and rough slopes are tested in the range $[-1.0,\, 1.0] \,\text{m/s} \times [-1.0,\, 1.0] \,\text{m/s} \times [-1.0,\, 1.0] \,\text{rad/s}$ and $[-2.0,\, 2.0] \,\text{m/s} \times [-1.0,\, 1.0] \,\text{m/s} \times [-2.0,\, 2.0] \,\text{rads/s}$. All other tests are conducted within the range $[-1.0,\, 1.0] \,\text{m/s} \times [-1.0,\, 1.0] \,\text{m/s} \times [-1.0,\, 1.0] \,\text{rad/s}$. These ranges correspond to linear velocity in the forward direction, lateral directions, and angular velocity about the vertical axis, respectively.
>
> When testing the linear velocity tracking error, the angular velocity target is set to 0. For the angular velocity tracking error tests, the linear velocity target is 0. The 'combined velocity' tests are conducted without setting either velocity component to 0. We uniformly sample test commands within these ranges. The environmental conditions are set to $\text{level}^{\text{terrain}} = 1, 2, 3, 4$, with an equal distribution of 25\mathrm{percent} for each. $\text{level}^{\text{terrain}} = 1, 2, 3, 4$ means that the slopes and rough slopes have a inclination of 4.44, 8.89, 13.33, 17.78 degree respectively, the rough slopes also have uniform noise of amplitude 1.78, 2.56, 3.33, 4.11 cm, the stairs' heights are 7, 9, 11, 13 cm respectively, the discrete obstacles' heights are 6.11, 7.22, 8.33, 9.44 cm respectively.
>
> The compared methods include ours, regression version of ours, baseline, RMA[1] and MoB[2], the results are shown in the following table:
> | Terrain | Velocity | Range | Ours  | Regression | RMA   | Baseline | MoB (Trot) |
> |  :---:  |   :---:  | :---: | :---: |    :---:   | :---: | :---:    |    :---:   |
> | Slopes | Linear | $[-1,1] \mathrm{m} / \mathrm{s}$ | 0.073 | **0.071** | 0.158 | 0.271 | 0.103 |
> |  |              | $[-2,2] \mathrm{m} / \mathrm{s}$ | **0.075** | 0.077 | 0.270 | 0.305 | 0.126 |
> |  | Angular | $[-1,1] \mathrm{rad} / \mathrm{s}$ | 0.084 | 0.086 | 0.091 | 0.164 | **0.081** |
> |  |  | $[-2,2] \mathrm{rad} / \mathrm{s}$ | 0.124 | 0.130 | 0.134 | 0.202 | **0.114** |
> |  | Combined | $[-1,1] \mathrm{m} / \mathrm{s}$ | 0.078 | 0.075 | 0.155 | 0.207 | **0.069** |
> |  |  | $[-1,1] \mathrm{rad} / \mathrm{s}$ | **0.047** | 0.050 | 0.074 | 0.143 | 0.052 |
> |  | Combined | $[-2,2] \mathrm{m} / \mathrm{s}$ | **0.094** | 0.107 | 0.203 | 0.224 | 0.103 |
> |  |  | $[-2,2] \mathrm{rad} / \mathrm{s}$ | **0.067** | 0.071 | 0.102 | 0.157 | 0.069 |
> | Rough  slopes | Linear | $[-1,1] \mathrm{m} / \mathrm{s}$ | **0.086** | 0.091 | 0.201 | 0.287 | 0.105 |
> |  |  | $[-2,2] \mathrm{m} / \mathrm{s}$ | **0.085** | 0.088 | 0.294 | 0.331 | 0.124 |
> |  | Angular | $[-1,1] \mathrm{rad} / \mathrm{s}$ | **0.097** | 0.108 | 0.142 | 0.160 | 0.119 |
> |  |  | $[-2,2] \mathrm{rad} / \mathrm{s}$ | 0.143 | 0.150 | 0.147 | 0.187 | **0.135** |
> |  | Combined | $[-1,1] \mathrm{m} / \mathrm{s}$ | **0.088** | 0.095 | 0.167 | 0.220 | 0.147 |
> |  |  | $[-1,1] \mathrm{rad} / \mathrm{s}$ | **0.054** | 0.070 | 0.086 | 0.169 | 0.092 |
> |  | Combined | $[-2,2] \mathrm{m} / \mathrm{s}$ | **0.103** | 0.137 | 0.221 | 0.284 | 0.195 |
> |  |  | $[-2,2] \mathrm{rad} / \mathrm{s}$ | **0.076** | 0.096 | 0.117 | 0.201 | 0.114 |
> | Stairs | Linear | $[-1,1] \mathrm{m} / \mathrm{s}$ | **0.147** | 0.194 | 0.306 | 0.377 | 1.108 |
> |  | Angular | $[-1,1] \mathrm{rad} / \mathrm{s}$ | **0.087** | 0.101 | 0.241 | 0.370 | 0.930 |
> |  | Combined | $[-1,1] \mathrm{m} / \mathrm{s}$ | **0.135** | 0.217 | 0.337 | 0.405 | 1.270 |
> |  |  | $[-1,1] \mathrm{rad} / \mathrm{s}$ | **0.096** | 0.154 | 0.201 | 0.289 | 0.992 |
> | Discretre  obstacles | Linear | $[-1,1] \mathrm{m} / \mathrm{s}$ | **0.125** | 0.182 | 0.311 | 0.279 | 0.973 |
> |  | Angular | $[-1,1] \mathrm{rad} / \mathrm{s}$ | **0.083** | 0.097 | 0.175 | 0.253 | 0.841 |
> |  | Combined | $[-1,1] \mathrm{m} / \mathrm{s}$ | **0.125** | 0.206 | 0.305 | 0.374 | 1.254 |
> |  |  | $[-1,1] \mathrm{rad} / \mathrm{s}$ | **0.075** | 0.133 | 0.177 | 0.246 | 0.976 |
>
> The unit of tracking error for linear velocity and angular velocity are $m/s$ and $rad/s$ respectively. In most of the complex terrains, our method outperforms other methods sigificantly.
>
> ## Reference
> [1] Kumar, et al. RMA: Rapid Motor Adaptation for Legged Robots
>
> [2] Margolis, et al. Walk These Ways: Tuning Robot Control for Generalization with Multiplicity of Behavior

---

> ### Author Response · Authors · 2023-11-21
> **Extension of Experiment Part(Comparison part 2)**
>
> We also conduct real-world experiments to compare our method with RMA[1], MoB[2] and Built-in MPC, the detailed benchmark setting are in $\color{red}{appendix.2}$, which is a little long for comments here. The results show that our method have good performance in real world application:
>
> | Benchmarks | Environment | Metrics | Ours | RMA | MoB | Built-in MPC |
> | :---: | :---: | :---: | :---: | :---: | :---: | :---: |
> | Stairs | Short-range  (A1) | Success rate  $(\mathrm{percent})$ | **100** | 60 | 0 | 0 |
> |  | Long-range  (Aliengo) | Number  of stairs | **176.5 $\pm$ 7.81** | $75.35 \pm 19.98$ | $0.0 \pm 0.0$ | $0.0 \pm 0.0$ |
> | Unseen  terrains | Compositional  Terrain  (Aliengo) | Success rate  $(\mathrm{percent})$ | **85** | 45 | 0 | 0 |
> |  | Deformable Slope  (A1) | Success rate  $(\mathrm{percent})$ | **55** | 10 | 0 | 0 |
> | Anti-  disturbance | Dragging  Obstacle  (A1) | Maximum  weight  $(\mathrm{Kg})$ | **10** | **10** | 7 | 3 |
> |  | Vertical Hit  (A1) | Maximum  weight  $(\mathrm{Kg})$ | **8** | **7.5** | 7 | 5 |
> |  | Payload  (A1) | Maximum  weight  $(\mathrm{Kg})$ | **8** | 7 | 4 | 7 |
> |  | Missing steps  (Aliengo) | Success rate  $(\mathrm{percent})$ | **100** | 0 | 0 | 0 |
>
> ## Reference
> [1] Kumar, et al. RMA: Rapid Motor Adaptation for Legged Robots
>
> [2] Margolis, et al. Walk These Ways: Tuning Robot Control for Generalization with Multiplicity of Behavior

---

> > ### Author Response · Authors · 2023-11-21
> > **Extension of Experiment Part(Ablation part)**
> >
> > We also did extensive ablation studies ablation studies on  contrastive learning and regression. To be more specificly, we perform ablation study with the following methods:
> >
> > * **Baseline**: Training the policy with no auxiliary task, but with a history encoder of the same architecture training as our method, the history encoder is trained with policy update.
> > * **Ours w/o Velocity Input**: Training the policy the same as our method, but set the position of velocity input to 0 for the policy network.
> > * **Ours w/o Velocity Loss**: Training the policy the same as our method, but without the loss of velocity estimation and set the position of velocity input to 0 for the policy network.
> > * **Ours w/o Latent Input**: Training the policy the same as our method, but set the position of latent input to 0 for the policy network.
> > * **Ours w/o Latent Loss**: Training the policy the same as our method, but without the loss of dynamics-aware estimation and set the position of latent input to 0 for the policy network.
> > * **Regression**: Training policy with velocity estimation and dynamics-aware estimation, but the auxiliary is to simply regress them in latent space.
> > * **Oracle**: Training policy with a history of full observations.
> >
> > The training curve is in $\color{red}{section4.5}$, which can not be included in the comments due to the limitation of openreview. However, the real-world experiments(same as the comparision method) can show the improvement of our dynamics-aware modeling and contrastive learning method:
> > | Benchmarks | Environments | Metrics | Ours | Ours w/o  Vel. Inp. | Ours w/o  Vel. Loss | Ours w/o  Lat. Inp. | Ours w/o  Lat. Loss | Regression | Baseline |
> > | :---: | :---: | :---: | :---: | :---: | :---: | :---: | :---: | :---: | :---: |
> > | Stairs | Short-range  (A1) | Success rate  $(\mathrm{percent})$ | **100** | 85 | 80 | 50 | 55 | 85 | 10 |
> > |  | Long-range  (Aliengo) | Number  of stairs | **176.5 $\pm$ 7.81** | $51.45 \pm 25.67$ | $54.6 \pm 24.68$ | $13.3 \pm 13.42$ | $10.6 \pm 9.68$ | $155.7 \pm 13.48$ | $0.0 \pm 0.0$ |
> > | Unseen  terrains | Compositional  Terrain  (Aliengo) | Success rate  $(\mathrm{percent})$ | **85** | 70 | 75 | 50 | 50 | 70 | 15 |
> > |  | Deformable Slope  (A1) | Success rate  $(\mathrm{percent})$ | **55** | 30 | 25 | 0 | 0 | 30 | 0 |
> > | Anti-  disturbance | Dragging  Obstacle  (A1) | Maximum  weight  $(\mathrm{Kg})$ | **10** | **10** | **10** | 5 | 5 | **10** | 4 |
> > |  | Vertical Hit  (A1) | Maximum  weight  $(\mathrm{Kg})$ | **8** | 7 | 6.5 | 4 | 4 | **8** | 5.5 |
> > |  | Payload  (A1) | Maximum  weight  (Kg) | **8** | 7 | 7 | 3.5 | 3.5 | **7.5** | 3 |
> > |  | Missing steps  (Aliengo) | Success rate  $(\mathrm{percent})$ | **100** | 90 | 90 | 5 | 0 | 65 | 0 |

---

> > > ### Author Response · Authors · 2023-11-21
> > > **Response to Reviewer 9L23**
> > >
> > > In the remaining time we would like to address the remaining concerns that you have that prevent you from accepting the paper. We would appreciate it if you could engage with us on the remaining concerns, as we want to address them.

---

> > > > ### Comment · Reviewer_9L23 · 2023-11-22
> > > > **Response**
> > > >
> > > > Thank you authors for addressing many of the concerns and updating the draft, this work is in a much better shape now.
> > > > I'll raise my evaluation to 5 for now.

---

> > > > > ### Author Response · Authors · 2023-11-22
> > > > > **Response to Reviewer 9L23**
> > > > >
> > > > > We are delighted to hear that we have addressed some of your concerns! We thank you again for your invaluable feedback and insights. Meanwhile, we fully agree that the paper is in a much better position as a result of the discussion.
> > > > >
> > > > > We also want to share the insights of our work and some extra experiments we have done:
> > > > >
> > > > > ## Insights:
> > > > > * Previous works[1, 2, 3] use adaptation method or teacher&student, they try to perform like an agent with full observation. Our method directly treats different terrains as perturbations, and try to learn the best response under different observations with the help of critic module. A vivid analogy is that previous work trains a dog that can see the terrain and then blindfolded it to achieve prior performance, which is obviously impossible; the dog we trained has never seen the terrain, and learns to react directly to a variety of different situations.
> > > > > * To model the scene that a blind dog faces, we can not let the dog see what  exact the environment is. But we can let it sense that under current state, if no action will be perform, what will happen. For example, if the dog's front leg hits a stair, even if it can not see, it should know that raising its leg higher is a necessary condition of overcoming the unseen stairs. In our work, with this insight, we treat the external environment as a kind of system disturbance(the impact of friction, resitution, terrain, etc), then use the internal model to estimate such information. In fact, it comes from the idea of Internal Model Theory[4], which simulates the successor state of the robotic system in order to estimate the system disturbance brought by the environment.
> > > > >
> > > > > ## More Experiments:
> > > > > Besides the benchmarks and ablations, we also investigate the influence of prototype number. The number of prototypes in a way reflects the distribution of latent features in the latent space, the higher the number of prototypes, the more decentralized the latent features are. In the other aspect, the experiment proves that regression is not as good as contrastive learning in the dynamics-aware representation in locomotion task since regression always results in the decentralized distribution of latent features.
> > > > >
> > > > > We again thank you for your time and effort of reviewing our work. If you have any other concerns or suggestions, please let us know. We really enjoy the discussion with you!
> > > > >
> > > > > ## Reference
> > > > >
> > > > > [1] Kumar, et al. RMA: Rapid Motor Adaptation for Legged Robots
> > > > >
> > > > > [2] Lee, et al. Learning Quadrupedal Locomotion over Challenging Terrain
> > > > >
> > > > > [3] Miki, et al. Learning robust perceptive locomotion for quadrupedal robots in the wild
> > > > >
> > > > > [4] Rivera, et al. Internal model control: PID controller design

---

### Official Review · Reviewer_Yqja · 2023-10-29

**Soundness:** 3 good
**Presentation:** 3 good
**Contribution:** 3 good
**Rating:** 8
**Confidence:** 4

**Summary:**

This paper proposes to use a learned model to improve learning policies for a quadrupedal robot. In particular, the learned model is trained via contrastive learning instead of the typical regressive learning in existing literature.

**Strengths:**

1. Using contrastive learning for learning the dynamics model seems interesting.

2. Real robot experiments.

**Weaknesses:**

1. The paper seems to be written in a rush with lots of typos and incorrect references.

2. The need of the velocity model is a little bit unsatisfying as one would expect a perfect internal model should be able to replace it.

3. Need clarification for some questions that I will list below.

**Questions:**

1. Seems to me the proposed model is more of a state estimation model (or a state compression model) instead of a dynamics model. For example, one would expect a dynamics model to be used to do rollouts instead of using a simulator during training. It will be nice to have some comments about it.

2. A key missing ablations is to compare the proposed contrastive learning with a regression model.

3. It would be nice to have more analysis of the learned latent model as this is the key contribution. For example, examining the effect of the number of prototypes,

---

> ### Author Response · Authors · 2023-11-17
> **Response to Reviewer Yqja**
>
> # Response to Reviewer Yqja
>
> ## Comment:
> Thank you for your review and comments. To respond/comment on your listed weaknesses:
>
> W1: Thanks for your advice, we carefully checked the typos and incorrect reference in our paper, we have already corrected them in a the current version.
>
> W2: We have done some ablation study on the use of velicity estimation, while only using implicit internal model can already achieve satisfying performance, the experimental results show that using velicity estimation can improve the performance. However, the implicit internal model plays the main role of performance. You can refer to our new ablation study in $\color{red}{section 4.5}$ both in simulation and real-world.
>
> W3: We will answer the questions in the following contents.
>
> ## Questions
>
> Q1: Yes, our "dynamics model" is not the typical "dynamics model" or "world model" in Reinforcement Learning. We do not rollout on "dynamics model" to generate synthetic interaction data. Instead, the "dynamics model" is trained to help the policy understanding that under current state, if we do not take any action, how will the robot move. This in fact reflects the properties of the physical world, letting the policy determine how to react to achieve our command. More specifically, we treat the external environment as a kind of system disturbance(the impact of friction, resitution, terrain, etc), then use the "dynamics model" or "internal model" in our work to simulate the successor state of the system in order to estimate the potential disturbance. We think that "dynamics-aware model" should be a more accurate description, we already corrected this term in current version.
>
> Q2: We have done the ablation study between contrastive learning and regression, it shows that contrastive learning performs better, the superior performance comes from the utlization of batch level information and discriminative representation. Furthermore, we also conduct ablation studies on different variants of our method. You checkout the details in $\color{red}{section 4.5}$.
>
> Q3: Thank you for your advice on experiments, we conduct the ablation on the number of prototypes in $\color{red}{section 4.6}$, we also investigate the influence of velocity estimation(both input and auxiliary loss) and latent (both input and auxiliary loss) in $\color{red}{section 4.5}$, which shows that the latent(dynamics-aware) representaion plays a main role in the performance.

---

> > ### Comment · Reviewer_Yqja · 2023-11-20
> > **response**
> >
> > Thanks for the update. I will raise my score accordingly.

---

> > > ### Author Response · Authors · 2023-11-21
> > > **Response to reviewer Yqja.**
> > >
> > > We are very happy to hear that the reviewer has raised the score! We thank the reviewer again for the invaluable feedback and insights.

---

### Official Review · Reviewer_FQfg · 2023-11-02

**Soundness:** 3 good
**Presentation:** 2 fair
**Contribution:** 3 good
**Rating:** 6
**Confidence:** 3

**Summary:**

The paper proposed a new legged locomotion method called HIM (Hybrid Internal Model). The method uses only joint encoders and IMU observations (no cameras or range sensors). HIM has 2 modules, the "information extractor " which learns the environment dynamics, and the "policy network" which learns the policy. The HIM model is trained in a simulator (IsaacGym) and successfully deployed in the real world (Unitree Go1 robot).

**Strengths:**

The main strength of the paper is the proposal of the HIM (Hybrid Internal Model), which is able to address the legged locomotion problem using only joint encoders and IMU observations (no cameras or range sensors). This is achieved using two components, the "information extractor " which learns the environment dynamics, and the "policy network" which learns the policy. This is a very interesting approach and from my point of view is related to the "world models" concept.

The method worked successfully in different simulated and real environments. In the real world, experiments were conducted on various environments, showing agile and robust locomotion.

**Weaknesses:**

From my point of view "world models" also learn the dynamics of the environment, in simulation or without using simulators, and it is required that authors compares its proposal with world models. For instance, Wu et al. "Daydreamer: World models for physical robot learning" applies word models online and address legged locomotion.

This is important in order to better assess the originality and contributions of the paper.

**Questions:**

How your work is related with "world models" that also learn the dynamics of the environment? For instance, [1] applies word models online and also addresses the legged locomotion problem. I know that in your work less sensor data is used, but in addition to this, how your approach is compared with world models? Can your approach be considered a world model?

[1] Wu et al. "Daydreamer: World models for physical robot learning" applies word models online and address legged locomotion.

**Details Of Ethics Concerns:**

No ethical concerns.

---

> ### Author Response · Authors · 2023-11-17
> **Response to Reviewer FQfg**
>
> # Response to Reviewer FQfg
>
> ## Comment:
> Thank you for your review and comments. To respond/comment on your listed weaknesses:
>
> W1: The main difference between our work and "world models" is that we do not need the rollout process on "world models" to generate synthetic interaction data. In our work, the "dynamics model" is trained to help the policy understanding that under current state, if we do not take any action, how will the robot move. This in fact reflects the properties of the physical world, letting the policy determine how to react to achieve our command. More specifically, we treat the external environment properties(the impact of friction, resitution, terrain, etc) as a kind of disturbance, then use the "dynamics model" or "internal model" in our work to simulate the successor state of the system in order to estimate the potential system disturbance. For clarifcation, we refer to use "dynamics-aware model" in current version.
>
> ## Questions
> Q1: The relation between our work and "world models" is described in the comment to weakness. Besides, world models take historical state and action as input, while our method takes only historical state as input. Therefore, our method can not be considered as a world model. In fact, it comes from the idea of Internal Model Theory[1], which simulates the successor state of the robotic system in order to estimate the system disturbance brought by the environment. We add our inspiration and the relation between our method and world model in $\color{red}{section 3.3}$.
>
> ## Reference
>
> [1] Rivera, et al. Internal model control: PID controller design

---

> > ### Author Response · Authors · 2023-11-21
> > **Response to Reviewer FQfg**
> >
> > In the remaining time we would like to address your remaining concerns. We would appreciate it if you could engage with us on the remaining concerns, as we want to address them.

---

> > > ### Author Response · Authors · 2023-11-23
> > > **Further Discussion**
> > >
> > > Dear reviewer FQfg:
> > >
> > > We would like to thank you again for your constructive comments and helpful suggestions. Since we are nearly at the end of the discussion phase(less than 7 hours left), we'd like to know whether our response has addressed your concerns and we are always looking forward to your further comments or suggestions.
> > >
> > > Best regards,
> > > Authors of 5999

---

### Official Review · Reviewer_SBNA · 2023-11-05

**Soundness:** 3 good
**Presentation:** 2 fair
**Contribution:** 2 fair
**Rating:** 6
**Confidence:** 4

**Summary:**

In this paper, the authors proposed a new auxiliary module, i.e. the hybrid internal model to help blind locomotion policy training. The hybrid internal model consumes past propiorceptive observation histories and produces a latent dynamic vector and estimated base body speed. Instead of using the regular regression loss, the latent dynamic vector is trained with a contrastive loss. The authors zero-shot sim2real transfer their policy to a Unitree robot and show that  their policy can outperform other blind baselines on a few difficult tasks including walking up/down large stairs and on soft terrains.

**Strengths:**

1) The paper shows great sim2real results. The learned gait looks very smooth, and more importantly, can solve difficult multi-terrains without using perception
2) Some novelties in introducing a contrastively learned environment/internal dynamics model.

**Weaknesses:**

1) There isn't a big delta from previous works, i.e. the ETH 2020 science paper also demonstrates similar capabilities.
2) While it is applaud to see good results on blind locomotion walking, without perception the policy is still fundamentally limited on difficult terrains. For example, the policy has to hit the vertical edges to sense the terrain before the hybrid internal model can "sense" the change in the environment. And there are already good perceptive locomotion works there.
3) The experiment result section of the paper is thin. For example, there is no quantitative results on success rate etc.
4) In the ablation study, the oracle policy seems learning slower than the proposed method, which is surprising.
5) Consider publishing the code with the paper. Right now there is a supplementary pdf which is a duplication of the main paper.

**Questions:**

Is the hybrid internal model co-trained with the policy or separately?

---

> ### Author Response · Authors · 2023-11-17
> **Response to Reviewer SBNA**
>
> # Response to Reviewer SBNA
> ## Comment:
> Thank you for your review and comments. To respond/comment on your listed weaknesses:
>
> W1: Previous works[1, 2, 3] use a mimic learning framework to predict unavailable terrain information in latent space, which entails additional performance reductions.Our method directly treats different terrains as perturbations, and try to learn the best response under different observations with the help of critic module, which is able to sense terrain information. A vivid analogy is that previous work trains a dog that can see the terrain and then blindfolded it to achieve prior performance; the dog we trained has never seen the terrain, and learns to react directly to a variety of different situations.
>
> W2: Perceptive locomotion require to utilize a wealth of sensor information, such as depth cameras, ToF cameras or LIDAR, to capture height information about the surrounding terrain, which plays a crucial role. However, most of the sensors mentioned above are very expensive and fragile, and the vast majority of quadruped robots are not equipped with these sensors and can only use feedback information from a basic IMU and motors. For example, Mini-cheetah only equips IMU and joint encoder, and Unitree Go1's default depth camera is hard to tackle. Besides, our policy do not always hit the vertical edge, in most of the environments, it spontaneously use its front leg to sense the front terrain, the videos in demo shows this strategy.
>
> W3: We are very thankful for your advice on experiments, we have already conducted extensive experiments in current version, including simulation benchmarks $\color{red}{section 4.3}$, real-world benchmarks $\color{red}{section 4.4}$, ablation study $\color{red}{section 4.5}$ and influence of prototypes $\color{red}{section 4.6}$.
>
> W4: We are sorry for the wrong setting of oracle, we only include the current oracle observation in the policy input rather than a history of oracle observations, which may decrease the performance and learning speed of oracle policy, the new curve is also included in the paper $\color{red}{section 4.5}$.
>
> W5: We are working on refine and adding comments to our code, we will try our best to release the code as soon as possible. In the current revision, we add our training hyper parameters in the appendix, which can be used directly for training.
>
>
> ## Questions
>
> Q1: The hybrid internal model is co-trained with the policy. In each iteration, we first collect trajectories, then train the hybrid intrernal model with the proposed method, after that, we froze the hybrid internal model and update the actor-critic part. We refined the describtion in our paper $\color{red}{section 3.5}$
>
> ## Reference
>
> [1] Kumar, et al. RMA: Rapid Motor Adaptation for Legged Robots
>
> [2] Lee, et al. Learning Quadrupedal Locomotion over Challenging Terrain
>
> [3] Miki, et al. Learning robust perceptive locomotion for quadrupedal robots in the wild

---

> ### Author Response · Authors · 2023-11-21
> **Extension of Experiment Part(Comparison part 1)**
>
> In order to compare our method with other blind locomotion method, we conduct a tracking error($\color{red}{the \ Euclidean \ distance \ between \ real \ velocity \ and \ target \ velocity}$) benchmark in simulation. The tracking error on slopes and rough slopes are tested in the range $[-1.0,\, 1.0] \,\text{m/s} \times [-1.0,\, 1.0] \,\text{m/s} \times [-1.0,\, 1.0] \,\text{rad/s}$ and $[-2.0,\, 2.0] \,\text{m/s} \times [-1.0,\, 1.0] \,\text{m/s} \times [-2.0,\, 2.0] \,\text{rads/s}$. All other tests are conducted within the range $[-1.0,\, 1.0] \,\text{m/s} \times [-1.0,\, 1.0] \,\text{m/s} \times [-1.0,\, 1.0] \,\text{rad/s}$. These ranges correspond to linear velocity in the forward direction, lateral directions, and angular velocity about the vertical axis, respectively.
>
> When testing the linear velocity tracking error, the angular velocity target is set to 0. For the angular velocity tracking error tests, the linear velocity target is 0. The 'combined velocity' tests are conducted without setting either velocity component to 0. We uniformly sample test commands within these ranges. The environmental conditions are set to $\text{level}^{\text{terrain}} = 1, 2, 3, 4$, with an equal distribution of 25\% for each. $\text{level}^{\text{terrain}} = 1, 2, 3, 4$ means that the slopes and rough slopes have a inclination of 4.44, 8.89, 13.33, 17.78 degree respectively, the rough slopes also have uniform noise of amplitude 1.78, 2.56, 3.33, 4.11 cm, the stairs' heights are 7, 9, 11, 13 cm respectively, the discrete obstacles' heights are 6.11, 7.22, 8.33, 9.44 cm respectively.
>
> The compared methods include ours, regression version of ours, baseline, RMA[1] and MoB[2], the results are shown in the following table:
> | Terrain | Velocity | Range | Ours  | Regression | RMA   | Baseline | MoB (Trot) |
> |  :---:  |   :---:  | :---: | :---: |    :---:   | :---: | :---:    |    :---:   |
> | Slopes | Linear | $[-1,1] \mathrm{m} / \mathrm{s}$ | 0.073 | **0.071** | 0.158 | 0.271 | 0.103 |
> |  |              | $[-2,2] \mathrm{m} / \mathrm{s}$ | **0.075** | 0.077 | 0.270 | 0.305 | 0.126 |
> |  | Angular | $[-1,1] \mathrm{rad} / \mathrm{s}$ | 0.084 | 0.086 | 0.091 | 0.164 | **0.081** |
> |  |  | $[-2,2] \mathrm{rad} / \mathrm{s}$ | 0.124 | 0.130 | 0.134 | 0.202 | **0.114** |
> |  | Combined | $[-1,1] \mathrm{m} / \mathrm{s}$ | 0.078 | 0.075 | 0.155 | 0.207 | **0.069** |
> |  |  | $[-1,1] \mathrm{rad} / \mathrm{s}$ | **0.047** | 0.050 | 0.074 | 0.143 | 0.052 |
> |  | Combined | $[-2,2] \mathrm{m} / \mathrm{s}$ | **0.094** | 0.107 | 0.203 | 0.224 | 0.103 |
> |  |  | $[-2,2] \mathrm{rad} / \mathrm{s}$ | **0.067** | 0.071 | 0.102 | 0.157 | 0.069 |
> | Rough  slopes | Linear | $[-1,1] \mathrm{m} / \mathrm{s}$ | **0.086** | 0.091 | 0.201 | 0.287 | 0.105 |
> |  |  | $[-2,2] \mathrm{m} / \mathrm{s}$ | **0.085** | 0.088 | 0.294 | 0.331 | 0.124 |
> |  | Angular | $[-1,1] \mathrm{rad} / \mathrm{s}$ | **0.097** | 0.108 | 0.142 | 0.160 | 0.119 |
> |  |  | $[-2,2] \mathrm{rad} / \mathrm{s}$ | 0.143 | 0.150 | 0.147 | 0.187 | **0.135** |
> |  | Combined | $[-1,1] \mathrm{m} / \mathrm{s}$ | **0.088** | 0.095 | 0.167 | 0.220 | 0.147 |
> |  |  | $[-1,1] \mathrm{rad} / \mathrm{s}$ | **0.054** | 0.070 | 0.086 | 0.169 | 0.092 |
> |  | Combined | $[-2,2] \mathrm{m} / \mathrm{s}$ | **0.103** | 0.137 | 0.221 | 0.284 | 0.195 |
> |  |  | $[-2,2] \mathrm{rad} / \mathrm{s}$ | **0.076** | 0.096 | 0.117 | 0.201 | 0.114 |
> | Stairs | Linear | $[-1,1] \mathrm{m} / \mathrm{s}$ | **0.147** | 0.194 | 0.306 | 0.377 | 1.108 |
> |  | Angular | $[-1,1] \mathrm{rad} / \mathrm{s}$ | **0.087** | 0.101 | 0.241 | 0.370 | 0.930 |
> |  | Combined | $[-1,1] \mathrm{m} / \mathrm{s}$ | **0.135** | 0.217 | 0.337 | 0.405 | 1.270 |
> |  |  | $[-1,1] \mathrm{rad} / \mathrm{s}$ | **0.096** | 0.154 | 0.201 | 0.289 | 0.992 |
> | Discretre obstacles | Linear | $[-1,1] \mathrm{m} / \mathrm{s}$ | **0.125** | 0.182 | 0.311 | 0.279 | 0.973 |
> |  | Angular | $[-1,1] \mathrm{rad} / \mathrm{s}$ | **0.083** | 0.097 | 0.175 | 0.253 | 0.841 |
> |  | Combined | $[-1,1] \mathrm{m} / \mathrm{s}$ | **0.125** | 0.206 | 0.305 | 0.374 | 1.254 |
> |  |  | $[-1,1] \mathrm{rad} / \mathrm{s}$ | **0.075** | 0.133 | 0.177 | 0.246 | 0.976 |
>
> The unit of tracking error for linear velocity and angular velocity are $m/s$ and $rad/s$ respectively. In most of the complex terrains, our method outperforms other methods sigificantly.
>
> ## Reference
> [1] Kumar, et al. RMA: Rapid Motor Adaptation for Legged Robots
>
> [2] Margolis, et al. Walk These Ways: Tuning Robot Control for Generalization with Multiplicity of Behavior

---

> > ### Comment · Reviewer_SBNA · 2023-12-04
> > **Raised the score**
> >
> > After seeing more comprehensive experiment results I decided to raise my score.

---

> ### Author Response · Authors · 2023-11-21
> **Extension of Experiment Part(Comparison part 2)**
>
> We also conduct real-world experiments to compare our method with RMA[1], MoB[2] and Built-in MPC, the detailed benchmark setting are in $\color{red}{appendix.2}$, which is a little long for comments here. The results show that our method have good performance in real world application:
>
> | Benchmarks | Environment | Metrics | Ours | RMA | MoB | Built-in MPC |
> | :---: | :---: | :---: | :---: | :---: | :---: | :---: |
> | Stairs | Short-range  (A1) | Success rate  $(\mathrm{percent})$ | **100** | 60 | 0 | 0 |
> |  | Long-range  (Aliengo) | Number  of stairs | **176.5 $\pm$ 7.81** | $75.35 \pm 19.98$ | $0.0 \pm 0.0$ | $0.0 \pm 0.0$ |
> | Unseen  terrains | Compositional  Terrain  (Aliengo) | Success rate  $(\mathrm{percent})$ | **85** | 45 | 0 | 0 |
> |  | Deformable Slope  (A1) | Success rate  $(\mathrm{percent})$ | **55** | 10 | 0 | 0 |
> | Anti-  disturbance | Dragging  Obstacle  (A1) | Maximum  weight  $(\mathrm{Kg})$ | **10** | **10** | 7 | 3 |
> |  | Vertical Hit  (A1) | Maximum  weight  $(\mathrm{Kg})$ | **8** | **7.5** | 7 | 5 |
> |  | Payload  (A1) | Maximum  weight  $(\mathrm{Kg})$ | **8** | 7 | 4 | 7 |
> |  | Missing steps  (Aliengo) | Success rate  $(\mathrm{percent})$ | **100** | 0 | 0 | 0 |
>
> ## Reference
> [1] Kumar, et al. RMA: Rapid Motor Adaptation for Legged Robots
>
> [2] Margolis, et al. Walk These Ways: Tuning Robot Control for Generalization with Multiplicity of Behavior

---

> ### Author Response · Authors · 2023-11-21
> **Extension of Experiment Part(Ablation part)**
>
> The training curve is in $\color{red}{section4.5}$, which can not be included in the comments due to the limitation of openreview. However, the real-world experiments(same as the comparision method) can show the improvement of our dynamics-aware modeling and contrastive learning method:
> | Benchmarks | Environments | Metrics | Ours | Ours w/o  Vel. Inp. | Ours w/o  Vel. Loss | Ours w/o  Lat. Inp. | Ours w/o  Lat. Loss | Regression | Baseline |
> | :---: | :---: | :---: | :---: | :---: | :---: | :---: | :---: | :---: | :---: |
> | Stairs | Short-range  (A1) | Success rate  $(\mathrm{percent})$ | **100** | 85 | 80 | 50 | 55 | 85 | 10 |
> |  | Long-range  (Aliengo) | Number  of stairs | **176.5 $\pm$ 7.81** | $51.45 \pm 25.67$ | $54.6 \pm 24.68$ | $13.3 \pm 13.42$ | $10.6 \pm 9.68$ | $155.7 \pm 13.48$ | $0.0 \pm 0.0$ |
> | Unseen  terrains | Compositional  Terrain  (Aliengo) | Success rate  $(\mathrm{percent})$ | **85** | 70 | 75 | 50 | 50 | 70 | 15 |
> |  | Deformable Slope  (A1) | Success rate  $(\mathrm{percent})$ | **55** | 30 | 25 | 0 | 0 | 30 | 0 |
> | Anti-  disturbance | Dragging  Obstacle  (A1) | Maximum  weight  $(\mathrm{Kg})$ | **10** | **10** | **10** | 5 | 5 | **10** | 4 |
> |  | Vertical Hit  (A1) | Maximum  weight  $(\mathrm{Kg})$ | **8** | 7 | 6.5 | 4 | 4 | **8** | 5.5 |
> |  | Payload  (A1) | Maximum  weight  (Kg) | **8** | 7 | 7 | 3.5 | 3.5 | **7.5** | 3 |
> |  | Missing steps  (Aliengo) | Success rate  $(\mathrm{percent})$ | **100** | 90 | 90 | 5 | 0 | 65 | 0 |

---

> > ### Author Response · Authors · 2023-11-21
> > **Response to Reviewer SBNA**
> >
> > In the remaining time we would like to address the remaining concerns that you have that prevent you from accepting the paper. We would appreciate it if you could engage with us on the remaining concerns, as we want to address them.

---

> > > ### Author Response · Authors · 2023-11-23
> > > **Further Discussion**
> > >
> > > Dear reviewer SBNA:
> > >
> > > We would like to thank you again for your constructive comments and helpful suggestions. Since we are nearly at the end of the discussion phase(less than 7 hours left), we'd like to know whether our response has addressed your concerns and we are always looking forward to your further comments or suggestions.
> > >
> > > Best regards,
> > > Authors of 5999

---

### Author Response · Authors · 2023-11-20
**Common Response**

Dear ICLR Reviewers,

We would like to express our sincere gratitude for your time and effort in reviewing our paper. Your constructive feedback has been invaluable in enhancing the quality of our work.

We have carefully addressed each of your comments and provided detailed responses in our revised manuscript. Additionally, we have made updates to the paper to incorporate your suggestions, and these revisions are highlighted in $\color{red}{red}$ for your convenience.

We eagerly anticipate your continued feedback and hope that you find the updated manuscript to be satisfactory.

Thank you again for your commitment.

Best regards,
Authors of 5999

---

### Author Response · Authors · 2023-11-23
**Further Discussion**

Dear reviewers and AC:

We would like to thank you again for your constructive comments and helpful suggestions. Since we are nearly at the end of the discussion phase(less than 7 hours left), we'd like to know whether our response has addressed your concerns and we are always looking forward to your further comments or suggestions.

Best regards,

 Authors of 5999

---

### Meta-Review · Area_Chair_9spb · 2023-12-08

**Metareview:**

*Summary*: This work aims to address the limitation of existing learning-based locomotion control methods using minimal sensor input. To do so, this paper proposed the Hybrid Internal Model that learns the latent representation of velocity and implicit dynamics of the environment via contrastive learning. The proposed method is evaluated extensively in simulations and the real world with comparisons with baselines (e.g., teach-student framework, regression version of the proposed method).

*Strength*: (1) This work only needs joint encoders and IMU as sensory inputs. (2) Impressive real-world performance in the video. (3) Learning environmental/internal observation model using constrastive learning for legged locomotion is novel. The t-SNE plot shows that the proposed method indeed learned to disentangle different terrain types in the latent space.

*Weakness*: (1) The original writing was in a rush with many typos and technical inaccuracies. Although it has been improved significantly during the rebuttal phase, it still has not fully met the ICLR standard. (2) The proposed joint encoder + IMU setting has some fundamental limitations. Compared to vision-based solutions, it is purely reactive and can not plan or reason ahead of time. There exist many vision-based impressive locomotion controllers.

**Justification For Why Not Higher Score:**

See the weakness part.

**Justification For Why Not Lower Score:**

See the strength part.

---

### Decision · Program_Chairs · 2024-01-16

Accept (poster)